# Nonlinear Volterra Integrodifferential Equations from above on Unbounded Time Scales

Andrejs Reinfelds [1,*] and Shraddha Christian [2,*]

1 Institute of Mathematics and Computer Science, LV 1459 Riga, Latvia
2 Institute of Applied Mathematics, Riga Technical University, LV 1048 Riga Latvia
* Correspondence: reinf@latnet.lv (A.R.); shraddha-ramanbhai.christian@rtu.lv (S.C.)

**Abstract:** The paper is devoted to studying the existence, uniqueness and certain growth rates of solutions with certain implicit Volterra-type integrodifferential equations on unbounded from above time scales. We consider the case where the integrand is estimated by the Lipschitz type function with respect to the unknown variable. Lipschitz coefficient is an unbounded rd-function and the Banach fixed-point theorem at a functional space endowed with a suitable Bielecki-type norm.

**Keywords:** Volterra integrodifferential equations; time scales; Banach fixed-point theorem; existence and uniqueness of solutions

**MSC:** 34K42; 45D05; 45G10

## 1. Introduction

This paper introduces the more general findings on the existence and uniqueness of the solutions to implicit Volterra-type integrodifferential equations on time scales.

The theory of time scales analysis has been growing fast and has received a lot of interest. This theory was introduced by Stefan Hilger [1] in 1988, and unifies continuous and discrete calculus. For a general introduction to time scale calculus and dynamic equations on time scales, see the books [2,3] by Bohner et al. In addition, we refer to the books of Georgiev [4] and Adivar et al. [5] and the paper of Karpuz [6] and the references therein for complete and extensive studies on recent results on time scales.

Economics is an ideal discipline with enormous potential for rich and diverse applications of time scales; thus, recently, lots of attention has been devoted to this study. Time scale calculus allows for a consideration of a variety of situations in economics. Atici et al. [7] present a dynamic optimization problem from economics and construct a time scale model. Furthermore, integral equations for time scales could be very useful for modelling economic processes, for example, a Keynesian Cross model with "lagged" income [8,9]. Applications of time scales can be found in mathematical biology and electrical engineering, see ([2], pp. 15–16), Messina et al. [10] consider Kernack and McKendrick's age-of-infection epidemic model. In addition, Sikorska's paper [11] on the integral equation for compact time scales can be noted, where the corresponding integrand is considered in the Henstock–Kurzweil delta integral sense, and the paper of Georgiev [12] considers an Adomin decomposition method for Volterra integrodifferential equations for time scales.

Kulik et al. [13] studied the basic qualitative and quantitative results of nonlinear Volterra integral equations for time scales. They consider the case when the integrand is estimated by the Lipschitz type function with respect to the unknown variable and Lipschitz coefficient is constant

$$x(t) = f(t) + \int_{t_0}^{t} K(t, \tau, x(\tau)) \, \Delta\tau, \quad t_0, t \in I_{\mathbb{T}} = [t_0, +\infty) \cap \mathbb{T}. \tag{1}$$

Reinfelds et al. [14–16] generalized results [13] using the Lipschitz-type rd-continuous function $L(t)$ instead of the Lipschitz coefficient, which can be unbounded. This turns

out to be necessary when considering the integral equations on an unbounded time scale. B.G. Pachpatte [17,18] and D.B. Pachpatte [19–21] have studied more general Volterra-type integral and integrodifferential equations on $[t_0, +\infty)$ and on time scales in which the integrand is Lipschitz with a fixed Lipschitz constant

$$x^\Delta(t) = f(t, x(t), x^\Delta(t), \int_{t_0}^t K(t, \tau, x(\tau), x^\Delta(\tau)) \, \Delta\tau), \; t_0, t \in I_\mathbb{T}, \; x(t_0) = x_0. \tag{2}$$

Many integrodifferential equations can be reduced to Volterra-type integral equations. Motivated by the above results, in this paper we consider nonlinear Volterra integrodifferential equations from above on unbounded arbitrary time scale of the form

$$x^\Delta(t) = f(t) + \int_{t_0}^t K(t, \tau, x(\tau), x^\Delta(\tau)) \, \Delta\tau, \quad x(t_0) = x_0.$$

In Christian's Ph.D. thesis ([22], p. 52), the author has written ideas about how it is possible to reduce implicit Volterra integrodifferential equations to Volterra integral equations. So, we introduce a new function:

$$y(t) = x^\Delta(t).$$

We have a system of Volterra integral equations

$$x(t) = x_0 + \int_{t_0}^t y(\tau) \, \Delta\tau,$$

$$y(t) = f(t) + \int_{t_0}^t K(t, \tau, x(\tau), y(\tau)) \, \Delta\tau.$$

We define a new function $z(t) = (x(t), y(t))$, where

$$F(t) = (x_0, f(t)), \quad k_1(t, \tau, z(\tau)) = (y(\tau), K(t, \tau, x(\tau), y(\tau))).$$

The system of the Volterra integral equation can be rewritten as follows:

$$z(t) = F(t) + \int_{t_0}^t k_1(t, \tau, z(\tau)) \, \Delta\tau, \quad t_0, t \in I_\mathbb{T} = [t_0, +\infty) \cap \mathbb{T}, \tag{3}$$

where $z \colon I_\mathbb{T} \to \mathbb{R}^{2n}$ is the unknown function to be found, $F \colon I_\mathbb{T} \to \mathbb{R}^{2n}$ and $k_1 \colon I_\mathbb{T} \times I_\mathbb{T} \times \mathbb{R}^{2n} \to \mathbb{R}^{2n}$ are given nonlinear functions. Equation (3) is known as a nonlinear Volterra integral equation on arbitrary time scales.

## 2. Notations and Preliminaries

The theory of time scales unifies continuous and discrete analysis. Let us define the time scales $\mathbb{T}$.

A time scale $\mathbb{T}$ is an arbitrary nonempty closed subset of the real numbers $\mathbb{R}$. We assume that any time scale $\mathbb{T}$ has the topology that it inherits from the standard topology on the real numbers $\mathbb{R}$. The most well-known examples of time scales are real numbers $\mathbb{R}$, the integers $\mathbb{Z}$, the natural numbers $\mathbb{N}$, the union of closed intervals $[0, 1] \cup [2, 3]$, $[0, 1] \cup \mathbb{N}$, quantum numbers $\overline{q^\mathbb{Z}}$ and Cantor set.

Since a time scale may or may not be connected, we need the concept of the jump operators to describe the structure of the time scale under consideration and also to define the delta derivative. The forward and the backward jump operators $\sigma, \rho \colon \mathbb{T} \to \mathbb{T}$ and the graininess $\mu \colon \mathbb{T} \to [0, +\infty)$ are defined, respectively, by

$\sigma(t) = \inf\{s \in \mathbb{T} \mid s > t\}, \quad \rho(t) = \sup\{s \in \mathbb{T} \mid s < t\}, \quad \mu(t) = \sigma(t) - t.$

The jump operators allow the classification of points in a time scale $\mathbb{T}$. If $\sigma(t) > t$, then the point $t \in \mathbb{T}$ is called right scattered, while if $\rho(t) < t$, then the point $t \in \mathbb{T}$ is called left scattered. If $\sigma(t) = t$, then $t \in \mathbb{T}$ is called right dense, while if $\rho(t) = t$, then $t \in \mathbb{T}$ is called left dense.

We say that $f\colon \mathbb{T} \to \mathbb{R}$ is rd-continuous provided f is continuous at each right-dense point of $\mathbb{T}$ and has a finite left-sided limit at each left-dense point of $\mathbb{T}$ and will be denoted by $C_{rd}$. The function $f\colon \mathbb{T} \to \mathbb{R}$ is regressive if

$$1 + \mu(t)f(t) \neq 0 \quad \text{for all} \quad t \in \mathbb{T}.$$

Fix $t \in \mathbb{T}^{\kappa}$ and let $x\colon \mathbb{T} \to \mathbb{R}$. The delta derivative (also Hilger derivative) $x^{\Delta}(t)$ exists if for every $\varepsilon > 0$ there exists a neighbourhood $N = (t - \delta, t + \delta) \cap \mathbb{T}$ for some $\delta > 0$ such that,

$$\left| (x(\sigma(t)) - x(s)) - x^{\Delta}(t)(\sigma(t) - s) \right| \leq \varepsilon |\sigma(t) - s| \text{ for all } s \in N.$$

Take $\mathbb{T} = \mathbb{R}$ and $x$ as differentiable in the ordinary sense at $t \in \mathbb{T}$, then $x^{\Delta}(t) = x'(t)$ is the derivative used in standard calculus. Take $\mathbb{T} = \mathbb{Z}$, then $x^{\Delta}(t) = \Delta x(t)$ is the forward difference operator used in the difference equation.

A function $F\colon \mathbb{T} \to \mathbb{R}$ is called an anti-derivative of $f\colon \mathbb{T} \to \mathbb{R}$ provided $F^{\Delta}(t) = f(t)$ holds for all $t \in I_{\mathbb{T}}$. In this case, we define the (Cauchy) delta integral of $f$ by

$$\int_r^s f(\tau)\,\Delta\tau = F(s) - F(r) \text{ for all } r, s \in \mathbb{T}.$$

If $\mathbb{T} = \mathbb{R}$, then

$$\int_r^s f(\tau)\,\Delta\tau = \int_r^s f(\tau)\,d\tau$$

while if $\mathbb{T} = \mathbb{Z}$, then

$$\int_r^s f(\tau)\,\Delta\tau = \sum_{\tau=r}^{s-1} f(\tau), \text{ if } r, s \in \mathbb{Z} \text{ and } r < s.$$

### 3. Existence and Uniqueness

Let $p\colon \mathbb{T} \to \mathbb{R}$ be a non-negative (and therefore regressive) and rd-continuous scalar function. The Cauchy initial value problem for the scalar linear equation

$$x^{\Delta}(t) = p(t)x(t), \quad x(t_0) = 1$$

has the unique solution $e_p(\cdot, t_0)\colon \mathbb{T} \to \mathbb{R}$ [2]. More explicitly, using the cylinder transformation, the exponential function $e_p(\cdot, t_0)$ is given by

$$e_p(t, t_0) = \exp\left( \int_{t_0}^t \xi_{\mu(\tau)}(p(\tau))\,\Delta\tau \right),$$

here, $\xi_h(\omega)$ is considered as

$$\xi_h(\omega) = \begin{cases} \omega, & h = 0 \\ \frac{1}{h}\log(1 + h\omega), & h > 0 \end{cases}$$

where log is a principle logarithm function.

Observe that we have Bernoulli's type inequality [23]

$$1 + \int_{t_0}^t p(\tau)\,\Delta\tau \leq e_p(t, t_0) \leq \exp\left( \int_{t_0}^t p(\tau)\,\Delta\tau \right) \tag{4}$$

for all $t \in I_{\mathbb{T}} = [t_0, +\infty) \cap \mathbb{T}$.

Let $|\cdot|$ denote the Euclidean norm on $\mathbb{R}^n$. We will consider the linear space of rd-continuous functions, such that

$$\sup_{t \in I_{\mathbb{T}}} \frac{\max(|x(t)|, |x^{\Delta}(t)|)}{e_p(t, t_0)} < \infty$$

and denote this special space by $C_p^1(I_{\mathbb{T}}; \mathbb{R}^{2n})$. The space $C_p^1(I_{\mathbb{T}}; \mathbb{R}^{2n})$ endowed with a Bielecki-type norm

$$\|z\|_p^1 = \sup_{t \in I_{\mathbb{T}}} \frac{\max(|x(t)|, |y(t)|)}{e_p(t, t_0)} = \sup_{t \in I_{\mathbb{T}}} \frac{\max(|x(t)|, |x^{\Delta}(t)|)}{e_p(t, t_0)}$$

is Banach space.

We generalize and at the same time simplify the results [13,17,18,20,21] assuming $L$ can be an unbounded rd-continuous function.

**Theorem 1.** *Consider the integral Equation* (3) *with* $I_{\mathbb{T}} = [t_0, \infty)_{\mathbb{T}}$. *Let* $k_1 \colon I_{\mathbb{T}} \times I_{\mathbb{T}} \times \mathbb{R}^{2n} \to \mathbb{R}^{2n}$ *be rd-continuous in its first and second variable,* $F \colon I_{\mathbb{T}} \to \mathbb{R}^{2n}$ *and* $L \colon I_{\mathbb{T}} \to \mathbb{R}$ *be rd-continuous,* $\beta > 1$ *and* $p = L(\tau)\beta$. *If*

$$|k_1(t, \tau, z) - k_1(t, \tau, z_1)| \le L(\tau)|z - z_1|, \quad z, z_1 \in \mathbb{R}^{2n}, \quad \tau < t, \tag{5}$$

$$n = \sup_{t \in I_{\mathbb{T}}} \frac{1}{e_p(t, t_0)} \left| F(t) + \int_a^t k_1(t, \tau, 0) \, \Delta\tau \right| < \infty \tag{6}$$

*then the integral Equation* (3) *has a unique solution* $z \in C_p^1(I_{\mathbb{T}}; \mathbb{R}^{2n})$.

**Proof.** Consider the following equivalent formulation of (3), namely,

$$z(t) = \left( F(t) + \int_{t_0}^t k_1(t, \tau, 0) \, \Delta\tau \right) + \int_{t_0}^t (k_1(t, \tau, z(\tau)) - k_1(t, \tau, 0)) \, \Delta\tau, \quad t \in I_{\mathbb{T}}. \tag{7}$$

We will show that (7) has a unique solution and thus (3) must also have a unique solution. Consider the complete metric space $C_p^1(I_{\mathbb{T}}; \mathbb{R}^{2n})$ and let $H$ be defined by

$$[Hz](t) = \left( F(t) + \int_{t_0}^t k_1(t, \tau, 0) \, \Delta\tau \right) + \int_{t_0}^t (k_1(t, \tau, z(\tau)) - k_1(t, \tau, 0)) \, \Delta\tau, \quad t \in I_{\mathbb{T}}. \tag{8}$$

The fixed point of $H$ will be a solution to (7). Thus, we want to prove that there exists a unique z, such that $Fz = z$. To do this, we show that the conditions of Banach's theorem are satisfied. Now, we show that $H \colon C_p^1(I_{\mathbb{T}}; \mathbb{R}^{2n}) \to C_p^1(I_{\mathbb{T}}; \mathbb{R}^{2n})$. Let $z \in C_p^1(I_{\mathbb{T}}; \mathbb{R}^{2n})$. Taking the norms in (8), we obtain

$$
\begin{aligned}
\|Hz\|_p^1 &= \sup_{t \in I_{\mathbb{T}}} \frac{1}{e_p(t, t_0)} \left| F(t) + \int_{t_0}^t k_1(t, \tau, 0) \, \Delta\tau + \int_{t_0}^t (k_1(t, \tau, z(\tau)) - k_1(t, \tau, 0)) \Delta\tau \right| \\
&\le n + \sup_{t \in I_{\mathbb{T}}} \frac{1}{e_p(t, t_0)} \int_{t_0}^t |k_1(t, \tau, z(\tau)) - k_1(t, \tau, 0)| \, \Delta\tau \\
&\le n + \sup_{t \in I_{\mathbb{T}}} \frac{1}{e_p(t, t_0)} \int_{t_0}^t L(\tau)|z(\tau)| \, \Delta\tau \\
&= n + \sup_{t \in I_{\mathbb{T}}} \frac{1}{e_p(t, t_0)} \int_{t_0}^t L(\tau)e_p(\tau, t_0) \frac{|z(\tau)|}{e_p(\tau, t_0)} \, \Delta\tau \\
&\le n + \|z\|_p^1 \sup_{t \in I_{\mathbb{T}}} \frac{1}{e_p(t, t_0)} \int_{t_0}^t L(\tau)e_p(\tau, t_0) \, \Delta\tau \\
&= n + \|z\|_p^1 \sup_{t \in I_{\mathbb{T}}} \frac{1}{e_p(t, t_0)} \frac{1}{\beta} \int_{t_0}^t \beta L(\tau)e_p(\tau, t_0) \, \Delta\tau \\
&= n + \|z\|_p^1 \sup_{t \in I_{\mathbb{T}}} \frac{1}{e_p(t, t_0)} \frac{1}{\beta} \int_a^t e_p^{\Delta}(\tau, t_0) \, \Delta\tau \\
&= n + \frac{\|z\|_p^1}{\beta} \sup_{t \in I_{\mathbb{T}}} \frac{1}{e_p(t, t_0)} [e_p(\tau, t_0)]_{t_0}^t
\end{aligned}
$$

$$
\begin{aligned}
&= n + \frac{\|z\|_p^1}{\beta} \sup_{t \in I_{\mathbb{T}}} \left( 1 - \frac{1}{e_p(t, t_0)} \right) \\
&\leq n + \frac{\|z\|_p^1}{\beta} < \infty
\end{aligned}
$$

This proves that the operator $H$ maps $C_p^1(I_{\mathbb{T}}; \mathbb{R}^{2n})$ into itself.

Next, we verify that $H$ is a contraction mapping with the contraction constant $\lambda = 1/\beta < 1$ and then Banach's fixed-point theorem will apply. For any $u, \overline{u} \in C_p^1(I_{\mathbb{T}}; \mathbb{R}^{2n})$

$$
\begin{aligned}
\|(Hu - H\overline{u})\|_p^1 &= \sup_{t \in I_{\mathbb{T}}} \frac{|[Hu](t) - [H\overline{u}](t)|}{e_p(t, t_0)} \\
&\leq \sup_{t \in I_{\mathbb{T}}} \frac{1}{e_p(t, t_0)} \int_{t_0}^t |k_1(t, \tau, u(\tau)) - k_1(t, \tau, \overline{u}(\tau))| \, \Delta\tau \\
&\leq \sup_{t \in I_{\mathbb{T}}} \frac{1}{e_p(t, t_0)} \int_{t_0}^t L(\tau) |u(\tau) - \overline{u}(\tau)| \, \Delta\tau \\
&= \sup_{t \in I_{\mathbb{T}}} \frac{1}{e_p(t, t_0)} \int_{t_0}^t L(\tau) e_p(\tau, t_0) \frac{|u(\tau) - \overline{u}(\tau)|}{e_p(\tau, t_0)} \, \Delta\tau \\
&\leq \|u - \overline{u}\|_p^1 \sup_{t \in I_{\mathbb{T}}} \frac{1}{e_p(t, t_0)} \int_{t_0}^t L(\tau) e_p(\tau, t_0) \, \Delta\tau \\
&= \|u - \overline{u}\|_p^1 \sup_{t \in I_{\mathbb{T}}} \frac{1}{e_p(t, t_0)} \frac{1}{\beta} \int_{t_0}^t \beta L(\tau) e_p(\tau, t_0) \, \Delta\tau \\
&= \frac{\|u - \overline{u}\|_p^1}{\beta} \sup_{t \in I_{\mathbb{T}}} \frac{1}{e_p(t, t_0)} \int_{t_0}^t e_p^{\Delta}(\tau, t_0) \, \Delta\tau \\
&= \frac{\|u - \overline{u}\|_p^1}{\beta} \sup_{t \in I_{\mathbb{T}}} \frac{1}{e_p(t, t_0)} [e_p(\tau, t_0)]_{t_0}^t \\
&= \frac{\|u - \overline{u}\|_p^1}{\beta} \sup_{t \in I_{\mathbb{T}}} \left( 1 - \frac{1}{e_p(t, t_0)} \right) \\
&\leq \frac{\|u - \overline{u}\|_p^1}{\beta} = \lambda \|u - \overline{u}\|_p^1.
\end{aligned}
$$

Since $\beta > 1$, it follows from the Banach fixed-point theorem with Bielecki-type norm [24] that $H$ has a unique fixed point $z$ in $C_p^1(I_{\mathbb{T}}; \mathbb{R}^{2n})$. The fixed point of $H$ is, however, a solution of (3). The proof is complete. $\square$

Theorem 1 also provides information about the behaviour of solution $z$ on the entire interval $I_{\mathbb{T}}$. We have certain growth rates of the solution:

$$
|z(t)| \leq M e_p(t, t_0) \leq M \exp\left( \int_{t_0}^t p(\tau) \, \Delta\tau \right), \quad M = \frac{n\beta}{\beta - 1}.
$$

**Example 1.** *Consider the nonlinear Volterra integrodifferential equation on $\mathbb{T}$*

$$
x^{\Delta}(t) = t^2 + \int_{t_0}^t (\tau + \sigma(\tau)) [x(\tau)^2 + x^{\Delta}(\tau)^2 + 1]^{\frac{1}{2}} \, \Delta\tau, \quad x(t_0) = x_0, \quad t_0, t \in I_{\mathbb{T}}, \quad t_0 \geq 1/2\sqrt{2}.
$$

*We will prove that this nonlinear Volterra integrodifferential equation has a unique solution for arbitrary unbounded time scales $\mathbb{T}$.*

**Proof.** We apply Theorem 1 and check the fact that $K(t, \tau, q, r) = (\tau + \sigma(\tau))(q^2 + r^2 + 1)^{\frac{1}{2}}$ has the bounded partial derivatives with respect to $q$ and $r$ everywhere and we have

$$|K(t, \tau, q, r) - K(t, \tau, q_1, r_1)| \leq \sqrt{2}(\tau + \sigma(\tau)) \max(|q - q_1|, |r - r_1|),$$

where we used Hadamard's lemma. So, (5) can be defined with $L(\tau) = \sqrt{2}(\tau + \sigma(\tau))$. We chose $\beta = \sqrt{2}$, then we have $p(\tau) = 2(\tau + \sigma(\tau))$, considering that

$$\int_{t_0}^{t} (\tau + \sigma(\tau)) \, \Delta\tau = t^2 - t_0^2$$

and $e_p(t, t_0) \geq 1 + 2(t^2 - t_0^2)$. We verified that (6) holds. The proof of the illustrative example now follows from Theorem 1. □

Using Pachpatte's results [17–21], it is not possible to prove the existence of the global solution of the Volterra integrodifferential equation under consideration.

## 4. Conclusions

In this article, we reduced the integrodifferential equation to the system of Volterra integral equations. It allows the theory of Volterra integral equations to be used. We use the rd-continuous function $L: I_{\mathbb{T}} \to \mathbb{R}$ instead of the Lipschitz coefficient. In our case, $L$ can be an unbounded function. In the proof, we used the characteristic properties of generalized exponential function for an arbitrary time scale $\mathbb{T}$. Such an approach assumes to obtain necessary conditions for the existence and uniqueness of the solutions to the nonlinear Volterra integrodifferential equations from above on unbounded time scales. Next, we have used a Banach space with a special type of Bielecki norm defined in it. Such an approach allows the use of the Banach contraction principle. Together, they make it possible to find a universal, and at the same time conditionally simple, proof for many basic properties of Volterra integrodifferential equations.

**Author Contributions:** All authors have equally contributed to each part of the manuscript. All authors have read and agreed to the published version of the manuscript.

**Funding:** This research is partially supported by the Institute of Mathematics and Computer Science University of Latvia. Project "Dynamic equations on time scales".

**Data Availability Statement:** This study does not make use of any data.

**Conflicts of Interest:** The authors declare no conflict of interest.

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
