# Peer review of "Nonlinear Volterra Integrodifferential Equations from above on Unbounded Time Scales"

_mathematics, doi:10.3390/math11071760_

Round 1

Reviewer 1 Report

I believe that the work presents a current and original theme, it is well written and the symbology is adequate. Although the results are new, I consider that the absence of a Conclusions section prevents a real assessment of the impact of this manuscript.

The work can be approved after correcting the aforementioned.

Author Response

  • We have changed the title of the article “Volterra Integrodifferential Equations on Unbounded Time Scales”.
  • We have revised the abstract, introduction and the list of references based on reviewers’ comments.
  • We have added conclusion & example.
  • D. B. Pachpatte investigated existence and uniqueness of solutions but he has not given conditions for the existence and uniqueness in the case when the time scales is unbounded as Lipschitz coefficient is unbounded. In this article, we have given sufficient conditions for the existence and uniqueness when Lipschitz coefficient is unbounded rd-function.

Reviewer 2 Report

1. A PhD thesis (2020) of one of the co-authors is available: https://dspace.lu.lv/dspace/bitstream/handle/7/53363/298-79000-Christian_Shraddha.Ramanbhai_sc16024.pdf?sequence=1

A link to this material with an indication of what will be new in this article in comparison with the thesis is strictly necessary.

Possibly, some results are given in Ref. [17] where the wrong year (2029) is indicated.

2. A number of other very important relevant links are missing, including an article with exactly the same title (!!!) [1]:

[1] Georgiev, Svetlin. (2017). Volterra Integro-Differential Equations on Time Scales. International Journal of Applied and Computational Mathematics. 3. 10.1007/s40819-016-0207-2. 

[2] Sikorska-Nowak, Aneta. (2011). Integro-differential equations on time scales with Henstock-Kurzweil delta integrals. Discussiones Mathematicae. Differential Inclusions, Control and Optimization. 31. 71. 10.7151/dmdico.1128. 

et al. ...

3. The introduction does not explicitly indicate the purpose and objectives of the work, research methods. The description of the problem is very short and difficult to understand. There is no detailed comparative analysis of other works in the field of integro-differential equations on time scales. The Conclusion section is missing. It is not clear how the results of the work can be useful in the future in theoretical and practical terms (for example, in modeling economic processes, as the authors mention in the Introduction).

The volume of the article is more suitable for a report at a conference.

Taking into account the aforementioned  remarks, and additionally the fact of the matching of titles of this manuscript and the work of S. Georgiev, the overall recommendation is to reject the article in the present form.

Author Response

  • We have changed the title of the article “Volterra Integrodifferential Equations on Unbounded Time Scales”.
  • We have revised the abstract, introduction and the list of references based on reviewers’ comments.
  • We have added conclusion & example.
  • https://dspace.lu.lv/dspace/bitstream/handle/7/53363/298-79000-Christian_Shraddha.Ramanbhai_sc16024.pdf?sequence=1  
  • In the Ph.D. thesis (2020), page no. 52, author has written only idea about how It is possible to reduce Volterra Integrodifferential equation to Volterra integral equations. However, in the article we proved it with Illustration.

Author Response

  • We have changed the title of the article “Volterra Integrodifferential Equations on Unbounded Time Scales”.
  • We have revised the abstract, introduction and the list of references based on reviewers’ comments.
  • We have added conclusion & example.
  • We have added some additional information about time scales with Illustration in the introduction.
  • About specific comments: (a & b) we made corrections, (c) Example is provided.

Reviewer 4 Report

The authors in this paper present some basic qualitative properties of a certain Volterra-type integrodifferential equation on time scales. 

In fact, the paper consists of one Theorem which generalize general result on existence and uniqueness for the solutions of Volterra-type integrodifferential equation on time scales. 

The result and its proof are correct, but they did not give any example to illustrate the importance of the result. So, the author should add examples to support their finding. 

Author Response

  • We have changed the title of the article “Volterra Integrodifferential Equations on Unbounded Time Scales”.
  • We have revised the abstract, introduction and the list of references based on reviewers’ comments.
  • We have added conclusion & example.

Round 2

Reviewer 2 Report

Accept in present form.

Reviewer 4 Report

The author has done all my comments.